# The Relationship between Gamified Physical Exercise and Mental Health in Adolescence: An Example of Open Innovation in Gamified Learning

**DOI:** 10.3390/healthcare12020124

**Published:** 2024-01-05

**Authors:** David Pérez-Jorge, María Carmen Martínez-Murciano, Ana Isabel Contreras-Madrid, Isabel Alonso-Rodríguez

**Affiliations:** 1Department of Didactics and Educational Research, Faculty of Education, University of La Laguna, 38200 La Laguna, Spain; mcarmen730@hotmail.com (M.C.M.-M.); ialonsor@ull.edu.es (I.A.-R.); 2Clinical Practice Unit, Department of Dentistry, Faculty of Health Sciences, University Fernando Pessoa Canarias, 35450 Las Palmas de Gran Canaria, Spain; acontreras@ufpcanarias.es

**Keywords:** gamification, mental health, physical activity, adolescence

## Abstract

Interest in gamified physical activity has been driven by its potential to benefit student mental health. Integrating gamified practices for mental health improvement represents a significant innovation within multidisciplinary approaches to enhancing mental well-being. This review follows the PRISMA (Preferred Reporting Items for Systematic Reviews and Meta-Analyses) guidelines and was conducted using the Scopus and Web of Science (WOS) databases, primary sources for education-related studies. Thirteen papers were analyzed, yielding important insights into the relationship between gamified physical activity and mental health. The findings indicate that gamified physical activity positively influences adolescents’ mental health and well-being. Additionally, there is a need for improved application and game design to enhance learning within school contexts. Tailoring exergames to fit specific disciplines and school-related characteristics can promote healthier mobile application usage and offer significant benefits for the mental health of young individuals. The difference between this study and previous ones is that it focuses on mobile applications for encouraging active living to improve quality of life and mental health.

## 1. Introduction

There are applications for performing gymnastics, such as “Zombies, Run!” and “Nike Training Club”, among others, which exemplify gamified exercise. These applications enhance physical well-being and play a crucial role in improving mental health. Regular engagement in such activities can lead to stress reduction, mood elevation, improved sleep quality, and increased self-esteem. Gamification in health and wellness applications is strategically utilized to make exercise more engaging, thereby aiding in the prevention and management of both physical and mental health issues, such as anxiety, depression, and cognitive deterioration. This approach underscores the integral relationship between physical activity and mental well-being.

According to [1], adolescence is a relevant period of development; it occurs between 10 and 19 years of age, and it is crucial for adequate mental and physical development, in which sports activity is essential for the benefits it implies. 

The authors of [2] highlighted the advantages of gamified sports practice related to feelings of well-being. The positive experiences derived from games mediated by virtual environments were highlighted by [3], especially that of good humor and decreases in negative feelings (which have been its main benefits). In [4], it was stated that gamification is a good practice for positive social support in the case of adolescents with some disease. 

Gamified virtual sports practice, such as running, swimming, and cycling, motivates and generates a more significant learning experience; moreover, gamified psychological strategy is currently being used to improve motivation for practicing these sports [5,6]. Now, various platforms and applications are available for gamified exercise, as exemplified by videogame consoles like Xbox with Kinect and Wii Fit with Balance Board [7]. These platforms transform physical exercise into a playful and entertaining experience, offering a unique way to combine physical activity with interactive gaming. These systems allow users to engage in various sports activities, from yoga to aerobics, in a fun and appealing manner, thus promoting an active and healthy lifestyle. Furthermore, children grow by playing; as such, gamification can be seen as something natural and consubstantial to this developmental period [8]. The purpose of gamification is to persuade users to change their habits or behavior through the fun of physical exercise [9].

As previously mentioned, various platforms and applications are available for gamified exercise, which transform physical activity into a playful and entertaining experience [7]. These tools make exercise more attractive and promote healthy habits, such as maintaining regularity in exercise, improving adherence to fitness programs, and developing healthy competition. Through gamification, greater social connection is encouraged, and self-regulation and personal progress monitoring are incentivized, increasing intrinsic motivation towards an active and healthy lifestyle.

Increasingly, gamification is gaining strength as an active methodology. Children like to play and be entertained; therefore, creating their own gamified apps [10,11,12] and carrying out playful, educational activities to learn in a fun way would be an interesting and innovative proposal. There is still a false belief that play has more to do with leisure than learning. However, the crucial role that the experience of pleasure has in the consolidation of learning [13,14,15], as well as in achieving a healthy and balanced development [16], has long been demonstrated. The author of [8] is an expert and thought leader in gamification. They ask parents to play with their children and understand the educational possibilities of videogames and the apps they frequently use.

Traditional video games favor the flow of positive feelings; however, gamification produces experiences loaded with feelings of power and autonomy that multiply the motivational [17,18] and emotional effects of the playful experience and consolidate behaviors or habits more effectively [9]. 

### 1.1. Mental Health Implications

The study of [1] refers to mental health problems and the extent to which they can cause risky behaviors for general health, as well as how they indirectly affect physical health. In this sense, there is no health if there is no mental health [19]. Mental illness brings with it economic, social, employability, and productivity problems, as well as critical repercussions for the families of those who are suffering from some mental health problem.

Suicide is the fourth leading cause of death among young people (i.e., those aged 15–29 years). Failure to address adolescent mental health disorders has consequences that extend into adulthood, with implications for physical and psychological health, as well as the possible prevention from leading fulfilling lives [1,20,21].

In [22], gamified physical practice was shown as an essential resource for preventing the onset of mental illness. Physical activity has shown to have a positive influence on physical and mental health; it increases well-being [23,24] as well as decreasing stress and depression [25,26].

A sedentary lifestyle, characterized by low or no physical activity, is known to be the fourth leading cause of death worldwide, and it represents a risk factor for heart disease, diabetes, and cancer [21]. A sedentary lifestyle is also a cause of early mortality [27], which is why the new global action plan on physical activity has established the goal of reducing physical inactivity by 10% in 2025 and 15% by 2030 [28]. Thus, the need arises to consider gamified physical activity as a possible solution, due to its potential motivational and positive effects on well-being, enjoyment, happiness, and fun [29].

In [30], it is stated that a gamification strategy increases motivation for physical activity, which itself decreases and prevents sedentarism [31]. Smartphones and health apps can help people to modify or maintain healthy behaviors, and the authors of [32,33] considered that gamification could promote healthy behaviors and habits.

In the last decade, there has been an increased interest in researching games related to healthy habits, possibly due to the diversification of their clinical application. The approach to researching gamification and health together from an interdisciplinary perspective is novel [34]. If gamification, as an innovative methodology, is applied to sports practice, then its benefits toward mental and physical health will increase the appreciation of the scope of these techniques.

### 1.2. Gamification in the Educational Context

Educational gamification appeared when elements based on game design were integrated into the design of the formative process [35]. Specifically, in Physical Education, gamifying means “transforming the class itself into a game based on a narrative” [36]. Didactic strategies and training environments that favor motivation and a classroom climate oriented to learning have been proposed [37,38]. These are fundamental conditions for activating the self-regulation cycles of learning [39,40], and, with it, the competence for autonomous and self-directed learning follows [41]. When this occurs, in this sense, the gamification of the process can create elements derived from the game experience, which can configure an actual learning situation [42]. This supposes not so much the stimulation of extrinsic motivation with prizes and incentives but rather the construction of a methodology that is part of a coherent and contextualized didactic proposal [43].

Information and communication technologies (ICT), social networks, and the development of specific apps focused on models for promoting individualized learning have departed from the model of single spaces (gymnasiums, physical education classrooms, etc.). This step can contribute to institutional improvement and change processes towards more inclusive institutional approaches adapted to diversity [44]. The literature evidences the benefits of gamification as a tool for educational inclusion and the convenience of structuring didactic proposals with this active methodology in the classroom. The aim of this is to achieve a greater involvement of teachers and students in the life of the classroom and the school [5,6].

In relation to non-formal and informal education contexts, the boom in portable technology, together with the value of games as a motivating element, have made the ways of staying active and the tools to achieve it more versatile [45].

Gamified physical activities are proving more attractive and exciting for young people and adolescents, who increasingly seem to report a greater interest in active living and in leisure models that are increasingly moving away from sedentary lifestyles [46,47]. Rewards, continuous updates, personalization of challenges and activities, and the duality of the modality (individual/group) favor and incentivize young people and adolescents to engage in physical activities [48]. Gamified physical activity benefits those who practice it, mainly because it improves the well-being, mental health, commitment, and satisfaction of those who perform it [49]. Gamification helps turn physical exercise into a habit [50], one of the basic principles of individual health engagement and care.

This study aims to determine the relationship and main implications of the practice of gamified physical activity and mental health in adolescents. 

Due to the absence of previous systematic reviews on the topic, all the articles written to date were considered. Bearing in mind that adolescence is a crucial stage in the development of individuals, we reviewed studies that had samples from those who were 10 to 19 years of age and who occupied part of primary and secondary compulsory and non-compulsory education.

We started from the idea that gamified physical exercise is a particularly effective technique for improving adolescents’ mental health.

The general aim of the study was to identify the effects of gamified physical exercise on SM. The specific objectives were as follows:To identify the effects of gamified physical activity on the mental health of adolescents.To assess the interest that the use of gamified physical exercise in adolescents arouses in the field of research.To identify the interest that gamified physical activity generates in adolescence.

## 2. Materials and Methods

Studies that focused on gamified physical activity as a strategy to promote mental health were analyzed. In addition, the studies considered were estimated for the case of adolescents; as such, they specifically focused on the last cycle of primary education and the second year of high school. The studies on these programs should be focused on the benefits and/or limitations of their implementation in relation to the development of mental health and well-being. At the beginning of this study, the criteria for the inclusion and exclusion of documents were established to carry out an adequate selection of sources (see Table 1).

The methodology used in this study was mixed and interpretative, and a systematic review of the scientific literature on the research topic was conducted following the PRISMA statement (Preferred Reporting Items for Systematic Reviews and Meta-Analyses). The existence of previous studies and the few reviews carried out on the topic [51] justify the review for the period estimated in this work. This modality of study, standard in the field of medical sciences, has emerged strongly in the field of education [52]. Thus, it has subsequently allowed for the confluence of both fields concerning the systematic analysis of the advances and main findings in relation to the benefits of gamified physical activity and its benefits for mental health.

The PRISMA statement is in itself a guide on the concepts and methodology considered during the development of systematic review studies [53]. The scientific literature on a topic is accessed to construct objective conclusions, whereby validity is achieved by showing evidence [54]. It should be noted that this study, in addition to providing knowledge on the proposed topics, also aims to mark the path for future lines of research that arise from reflecting on the status and scope of research on gamified sports practice and its benefits for the mental health of adolescents [55].

### 2.1. Review of the Selected Bibliography

To ensure the rigor and appropriateness of the studies considered, we identified the keywords that were most frequently used in the studies that have addressed the present line of work (gamified physical activity and mental health in adolescents). The Boolean OR and AND were used, and the combination used for the first search was Gamification AND “Mental health” OR “Physical exercise”. This first attempt resulted in an exceptionally high number of documents, but it also allowed for a more specific selection of words and search topics. The end of the primary education stage and the secondary and baccalaureate stages were considered, as they coincide with adolescence (a fundamental stage in the development of healthy habits and lifestyles in which sports activity and practice are especially important for students). The final search combination was Gamification AND (“Mental health” or “physical exercise”) AND (Adolescent OR Teen). 

### 2.2. Risk of Bias Analysis

The dimensions recommended by the Cochrane handbook for systematic reviews, which identifies five biases, were used to assess bias. This tool made it possible to analyze the existence of a low-risk or high risk bias based on the analysis carried out by three experts in the field. In the absence of complete information for analysis, the risk of bias was considered unclear. Thus, the selected documents were analyzed based on the abovementioned dimensions.

An analysis of the documents obtained was applied to estimate the degree of agreement and discrepancy between judges. The analysis was based on the inclusion and exclusion criteria established in addition to the requirements in the event of a tie. It should be noted that some of the agreements could be the result of chance, and that their random effect could give a more significant reliability effect than the real one [56]. Bennett’s coincidence coefficient was used to correct this effect. Its values range from 0 (no agreement) to 1 (perfect agreement), and the coincidence value was *S* = 0.92, which allowed us to estimate the final selection of documents as adequate.
$$S=\left(\frac{Fo}{N}-\frac{1}{K}\right)\cdot \left(\frac{K}{K-1}\right)$$

*Fo* = number of judge agreements.*N* = number of elements to be coded.*K* = total number of categories.

Observational bias refers to systematic distortion in the collection, analysis, or interpretation of observational data, which may influence the validity and reliability of the results. A moderate value bias was obtained, which allowed the sources to be considered and the analysis to continue.

### 2.3. Resources and Sources

The resources used for the information search strategy during the study were the two electronic databases of the Web of Science (WOS) and Scopus, which were accessed via the search engine of the library of the University of La Laguna. The documents were accessed from these sources since they are the primary databases that host the research on health and education. It was for this reason that both educational scientific search databases were selected.

## 3. Results

### Procedure

An exhaustive search was carried out from December 2022 to March 2023 using keywords and the same Boolean markers OR and AND as in the initial search. Inclusion criteria were then applied, thereby eliminating those documents that did not meet them. The bibliographic manager Mendeley was used to eliminate duplicate texts. After this, and through the reading of titles and abstracts by two independent judges, the studies that met the inclusion and exclusion criteria, in terms of investigating the use of gamified physical activity applications and their relationship with mental health in adolescents, were selected. Of the texts chosen, a complete reading was carried out and a final decision was made on the selection or rejection of the texts. After this, the content analysis of the fourteen selected studies was carried out. The analysis was carried out on the objectives, sample, methodology, main results, and conclusions of the studies. Thus, the main characteristics and formative effects of the practice of gamified physical activity were defined.

From the Scopus database, 241 papers were found for the combination Gamification and (“mental health” or “physical exercise”).

From the WOS database, 202 papers were found for the combination Gamification and (“mental health” or “physical exercise”).

A more specific search was conducted to focus on the educational stages of adolescence, and this was performed due to the fact that it is considered a crucial time in which the body and mind make changes, as well as due to gamified exercise potentially helping to maintain mental and physical health.

In Scopus, 53 articles were found for the combination Gamification and (“mental health” or “physical exercise”) and (School or education or “primary education” or “compulsory education” or “secondary education” or “university”).

In the WOS, 56 results were found for Gamification and (“mental health” or “physical exercise”) and (School or education or “primary education” or “compulsory education” or “secondary education” or “university”).

A total of 109 results were obtained, and 21 duplicates were eliminated. Inclusion and exclusion criteria were applied based on the title and abstract, and 42 articles were eliminated if they were in a language other than English or Spanish. The remaining 48 full texts were read, and 35 documents were discarded. The final sample of studies consisted of a total of 13 papers. A detailed search of the scientific literature is shown below, see Figure 1. 

## 4. Discussion

### 4.1. Characteristics of the Included Studies

We reviewed everything written on the subject thus far. The thirteen articles selected for the review were research studies published in English between the years 2017 and 2022, which ensured updated results on the use of gamified apps in physical activity in relation to the mental health of adolescents. The studies were mainly conducted in European countries (Spain, Belgium, Scotland, the United Kingdom, the Netherlands, and Austria) and also in China, the United Arab Emirates, Morocco, Australia, New Zealand, and the United States. The Spanish participation in 9 of the 15 studies was also particularly notable.

Three of the studies used a purely quantitative methodology (20%; N = 3), two used a qualitative methodology (13.3% N = 2), four were mixed studies (26.7%; N = 4), and six were systematic review studies on the use of apps in gamified physical activity (40% N = 6). certain assessment tools were commonly used in the studies: questionnaires (73.3%; N = 11), observations (13.3%; N = 2), interviews (26.7%; N = 4), focus groups (6.7%; N = 1), and observation scales (%, N = 1).

The most commonly used instruments were questionnaires followed by interviews and surveys. In most of the studies (60%; N = 9), the samples selected were adolescents, in which four were documented and two were apps. As for the results, all thirteen studies showed evidence of a positive relationship between gamified physical activity and improved well-being and mental health.

All the information can be seen in Table 2, which also lists the authorship, year of publication, purpose, sample, country, methodology with evaluation instrument, and the main results.

### 4.2. Analysis of the Selected Documents

An analysis of the documents was conducted in order to identify the mental and physical health domain to which gamification can be applied. They were then highlighted for their affinity and relevance to this study, and the result of this process is presented in the order that is shown in Table 2.

The study of [57] involved an analysis of the technical, educational, and psychological dimensions related to motivation and the stereotypes of the apps they studied. In the psychological dimension, motivation and the presence of stereotypes were evaluated. In terms of motivation, they measured whether there was gamification and rewards, whether there was access to training data, whether improvement was observed over time by monitoring progress, whether results were shared on social networks, and whether there was feedback. The game mechanics were extracted from each paper so as to identify those most frequently used in the healthcare sector. The applications investigated were those identified as the most widely used in the health sector, with game mechanics ranging from step counting, running, home and outdoor exercise, fitness, weight loss, height gain, hiking, and cycling. In terms of stereotypes, it was assessed whether gender-based behavioral patterns were used. The analysis of motivation showed that 48% of the apps scored good and excellent, and it also highlighted that the collective experience of network users was the activity that most motivated adolescents.

In order to define the stereotypes, predefined patterns of behavior were considered that indicated how women and men should be, act, think, and feel. No gender stereotypes were found in 58% of the apps, although only three (10%) were not considered as proposing gender roles for each exercise. Thirty-three percent of the apps analyzed showed few or quite a few stereotypes, and ten percent (three apps) showed many stereotypes. The authors considered that the apps aimed at weight loss were oriented to one of the genders, and this was evident even in the title.

They considered physical exercise for its aesthetic dimension without considering its relevance for the health, quality of life, and social relationships of the people who participate. Apps such as “Nike Run Club”, “Decathlon Coach-Fitness Run”, “Fitness Online-Exercises at home and in the gym”, and “Adidas Training” showed a commitment to minimizing stereotypes and prejudices related to physical exercise. There are apps for creating nature trails “Ko moot” and “AllTrails”, which allow for the creation of routes in groups; these apps can help to promote healthy exercise in the context of adolescents, although it was found that there are no specific apps on the market adapted to educational contexts. No specific apps were found for ages 12 to 17, 57% had no age restriction, 36% were classified as for ages 4 and over, 4% were for ages 9 and over, and 4% were for ages 17 and over. 

The study by the authors of [58], in their review of studies, found that active pedagogies favored the psychological and psychosocial aspects of students, thereby highlighting that gamification promotes success in students in the last years of primary education and the first years of secondary education, which is evidenced in their improvements in academic performance. The models developed by different authors that were used to create and build exercise applications were discussed. They explained that badges, points, rewards, and narrative were used in the documents. They evaluated them with different questionaries, some of them ad hoc, semi-structured, individual interviews; focus group interviews; and tests. Some students gave feedback about the fact that they preferred flipped classrooms rather than traditional classes. They noted that there are no studies in early childhood education, and that there are few in the stages prior to university; as such, they recommended further research. What was beyond doubt was that gamified practice in education improves motivation towards the subject of physical education and its practice in addition to improving cognitive performance. It was also found that gamification improved relational skills, autonomy, collaboration, and conflict resolution in primary and secondary school students.

In addition, the study of [59] analyzed 15 physical exercise and yoga apps, out of which six presented elements such as gamification, points, rewards, goals, or graphics. “Exercise and Yoga app for stress relief”, “Yoga for anxiety, stress and depression relief” and “Self-Management Depression: Daily Exercise (GGDE)” were some of the apps related to physical activity that were analyzed. These apps focus on problems such as anxiety, stress, depression, insomnia, and eating disorders. To manage these problems, they propose meditation, breathing exercises, mindfulness, and cognitive behavioral therapy. The results showed that 51% of the selected apps used gamification to motivate users to continue using them, and 32% provided social functions such as chats.

The study by [60] showed the main types of games tested and applied in the improvement of mental health. If more research, faster iterations, rapid testing, non-traditional collaborations, and user-centered approaches were not produced for responding to the diverse needs and preferences of users, in rapidly changing environments, gamification in health areas could be hampered. Regarding the relationship between physical activity and mental health, nine studies were found on exergames (games based on sport or movement, and whose use was mainly in the older adults tested). Significant effects on the improvement of depressive symptoms were reported.

In addition, the study of [61] conducted a physical gamified practice experience in a Scottish community. The intervention offered 20 local points of common interest, ‘Beat Boxes’, and a topic of conversation among the inhabitants of Stranraer of “the game itself”. The sample included people aged under 11 years old (out of 327, 12 completed); people aged 12–17 years old (out of 285, 15 completed); 18–29 years old (out of 216, 21 completed); and the rest were 858 people aged up to 70 years (out of which 99 completed). “Beat the Street” was used to increase user participation by designing gamified activities. Radio Frequency Identification (RFID) scanners, i.e., the ‘Beat Boxes’, were located half a mile apart throughout the city, and the residents received 10 points each time two consecutive ‘Beat Boxes’ were touched with an RFID card within 1 h. People competed to see which schools and groups conducted the most physical activity over the course of the game, with the highest scorers being rewarded. This study found a positive relationship between gamified physical activity and mental well-being.

The study by [62] used the free online program Lunar Magic School during the lockdown period of COVID 19 on students under 12 years of age and their families. This program was carried out for one month and included nine weekly activities based on physical exercise and music. Some of the examples of these activities were as follows: the creation of sports circuits or magic yoga, creative activities using household materials, emotional educational activities through drawing monsters and storytelling, etc. The gamification consisted of simulating a magic school in which each family could obtain medals as a reward for activities that were well performed.

Parents reduced their anxiety and perception of their children’s physical and psychological discomfort. The results have been encouraging, moving from risk scores due to the confinement situation to scores similar to those of the pre-pandemic period. The program helped the children to improve their emotional management, reduce their stress levels, and regain higher degrees of physical activity as a family.

The study by [63] developed an exergame platform to improve motivation levels toward sports practice from the age of 4 years. It analyzed the increase in depression and obesity in children, as well as its relationship with sedentary lifestyles and poor sports practice. This platform promotes self-regulation and autonomy in children. Competitiveness should be avoided, and collaboration should be favored. The social distance, lack of relational skills, and excessive time spent by children watching screens decrease their interest in sports practice, increasingly distancing these young people from a healthy model and lifestyle.

The study by the authors of [64] determined that the implementation of an intervention based on Teaching Personal and Social Responsibility (TPSR) and the use of gamified activity provided a positive emotional climate in the classroom. They used rewards and there was immediate and individual feedback on the students’ motor actions through a social network or face-to-face in physical education classes, all of which improved the cognitive performance of secondary school students. The intervention was able to improve some executive functions (EF) such as cognitive inhibition and verbal fluency. The observations of these data highlighted the importance of promoting and enhancing cognitive processes for better academic performance.

The study by [65] was conducted by applying the “Healthy Teens in School” program, which is a ten-week online program designed to promote a healthy lifestyle and reduce the risk of eating disorders and obesity. It uses gamification, assesses eating behavior and risk of its disorders, weight, weight/figure concerns, physical activity habits, stress management, depression, anxiety, self-esteem, and quality of life. A group of normative adolescents was assigned to the “Healthy Habits” track, and overweight adolescents were assigned to the “Weight Management” track. In ten modules, the students learned about building a healthy lifestyle, balanced eating, and physical activity habits, as well as about ways to improve their body image and satisfaction with their body.

The study by the authors of [63,66] dealt with the Healthy Jeart application, an app that seeks to encourage adolescents to adopt a healthy lifestyle based on physical exercise, adequate nutrition, and physical and psychological well-being. In addition, the app involves affective–sexual relationships, the use of ICT, and it also concerns itself with addictions. It is primarily focused on students between 8 and 16 years old and teachers. Avatar Jeart accompanies participants. Through the challenges that start in the gamified app and continue in the classroom, the authors created combinations through which to adapt the app to different educational stages. The app was created in 2008 and continues to be updated, thus aiming to keep the interest of its users. Students can create healthy ideas that have to be approved by the administrators before publication. Participants have to collect healthy food during the game and are encouraged to avoid unhealthy elements. There are different tips that refer to sport, physical activity, and sedentary lifestyles. “Food” is about disproving some of the myths and false beliefs related to food, as well as for suggesting healthy habits. “Physical well-being” deals with rest, sleep, personal hygiene, and time management. “Psychological wellness” deals with self-esteem, interpersonal skills, and emotional intelligence, and “Sexual affective” focuses on intimate relationships, delving into the myths of romantic love, and promoting healthy relationships. “Toxic and addictions” have content on alcohol, tobacco, cannabis, bongs, and other substances. “New technologies” deals with the good use and abuse of ICT, as well as how to protect oneself in social networks.

The study by [67] used an exergame (a digital motor game with the aim of stimulating motor skills in its users). Just Dance Now proved to be a suitable game for the practice of dance in educational centers. The learning content was gamified, and 10 exergame dances were selected from among the 300 dances on the web platform. The selection criteria were difficulty, motor skills, different cultural dances, and the appropriateness of values to the age of the students. The ClassDojo virtual platform was used to gamify the intervention sessions, and a game board was created with Microsoft Excel (v. 10). This study showed that gamification provided a greater overall positive feeling and more motivation in the majority of the student body. The exergames produced more fun and learning, as well as improving the coordination and motor skills of the students. Although scientific interest has been shown in understanding separate effects, the combination of gamification as a method and exergames as a tool is considered significant in terms of learning. In this case, the Mechanics–Dynamics–Aesthetics gamification model and the exergame Just Dance Now were used with significant benefits in improving motivation, well-being, and satisfaction.

In [68], it was found that most of the selected articles reported on the benefits of gamification and applied games that were specifically related to chronic disease rehabilitation, physical activity, and mental health. Gamification generates motivation and engagement in the short term. The time of that engagement must be increased for mental health and physical improvement by improving the apps, such that motivation and commitment to healthy lifestyles can be perpetuated.

In [69], it was shown that there is evidence for the effectiveness of physical activity gamification interventions in improving participation and engagement in physical activity. There is a need to evaluate the effectiveness of combining gamification with mobile activity devices in terms of promoting physical activity. The review revealed that the gamification of physical activity had been applied to various population groups and was widely distributed among young people, but it was less distributed among older adults and patients with any disease. Most of the studies (60%) combined gamification with mobile devices to enhance behavior changes towards physical activity, and 50% of the studies used theories or principles to design gamified physical activity interventions. The most commonly used game elements were goal setting, progress bars, rewards, points, and feedback. This study demonstrated that gamified interventions increase participation in physical activity; however, the results varied widely, and moderate changes were achieved.

## 5. Discussion

The main objective of this systematic review was to investigate the findings of studies on physical activity, gamified games, and mental health in adolescence. 

In [64], gamified sport practices were considered as being crucial for the development and activation of executive functions. This idea is shared with [70,71,72], in which it was stated that the relevance of executive functions in the establishment and preservation of physical and mental health to face risk factors such as stress, sleep disturbances, loneliness, or lack of exercise affects the development of these functions.

The benefits of physical activity for the maintenance of quality of life and a healthy mental state are results that have been evidenced in all the studies analyzed. Furthermore, coinciding with [3,4], psychologically positive interventions using virtual environments have been found. However, there are elements about which there are doubts regarding the extent of this effect. We refer to the variables that determine the actual achievement or value of the interventions, and these have to do with individual psychological aspects and motivation. The scope and consequences of body care are shown by this study; a good physical condition is required to favor good mental health. The positive effects analyzed are varied and multiple, some such as personal well-being, fun, sociability, satisfaction, motivation, autonomy, positive mood, and creativity, as well as decreases in feelings of shame, depression, and stress, were particular highlights. 

The authors of [57,61,62,68,72] found that physical activity produces mental well-being. The same opinion was detailed in [26] regarding exergames, and this was due to the fact that this type of game decreases levels of stress.

In addition, in [68], a gamified platform was created to prevent children from leading a sedentary life. In [46,47], it was considered that gamified physical activity prevents living a life with little or no physical exercise, and this was such because gymnastics was seen as a fun game and not as an obligation. When such a belief was present, its use increased in in adolescents.

Physical gamified practice favors sociability or socialization processes. In fact, studies such as those by the authors of [57,58,59,62,64,66,68] agree with the study described by the author of [73], who found better social functioning and good mental health in young adults. The author of [66] expressed concern about the toxic behaviors that can gestate in social networks. Another positive effect was noted by [65] when observing that a person feels satisfied with their own body when practicing exercise, especially gamified exercise. In this line, in [74], it was stated that a person who is satisfied with their life adopts a healthy approach that is modulated by sport practice, which leads to increasingly better mental and physical health.

Also, in the studies of [57,58,66], it was considered that gamified physical activity is motivating; however, in [68], it was indicated that motivation is ephemeral and that the effects are only short term, as there is a commitment to continue practicing physical activity. This is most likely subject to more reward-based approaches as a form of extrinsic motivation. In contrast, in the studies of [63,75], it was observed that the gamified practice of physical activity leads a person to engage continuously. This starts from a focus on the ability to integrate remote and face-to-face learning with the gamification supported by ICT to meet the basic psychological needs of young people, both in the classroom and outside physical education classes. This model generates self-motivation, physical literacy, and commitment to physical activity. The idea of such commitment has also been confirmed by the authors of [50], who see the model as encouraging continued physical activity.

In relation to the context of leisure and amusement, the studies of [29,62,66,69] considered that enjoyment favors good mental health, and, as stated by [76], assumed that the effect was therefore promoting the practice of physical activity through the game on cell phones. But, it is also crucial for professionals to advise their gamified practice in educational establishments [76,77].

Regarding the formal educational context, the studies of [78,79] found benefits for the mental health of primary school students in the practice of exergames. The practice of gamified fitness has brought benefits for the improvement of well-being in students of all educational stages, including the university level. 

The study of [58] agreed with that of [80] about the benefits of gamification in the teaching–learning process by considering it a determining factor of motivation towards learning. From an early age, people learn by playing, and it is logical to think that, at other educational stages, games are an element that favor the teaching and learning process.

In [62], it was considered that gamified physical activity decreases stress. This notion is similar to the one held by the authors of [26], who advised that exergames (i.e., the active video game form) also achieve this affection and improve the physical and cognitive abilities of students. 

In [58,63,69], the benefits of the development of autonomy and reduction of depressive symptoms, even among university students, was assessed. The authors of [60,65] shared with the authors of [59] the need to boost the design of physical activity apps for the benefits they provide to prevent depression. This could result in increasing the investigation to achieve a long-term motivating effect in these apps [81]. If this occurs, it will encourage the creation of new apps that help mental well-being. 

As a final synthesis, we can highlight the importance of physical activity in all educational stages, the benefits of exergames, and the gamification of activities, as well as highlight the value of this type of strategy for improving student motivation [18,82,83,84]. The benefits of these practices on physical and mental health, highlighted in all of the studies analyzed, require an effort to reinterpret the working model of physical activity in schools. The design of specific apps for the development of the formative process in educational institutions is a challenge that administrations ought to assume urgently. The professionals who design them could do so in a more specific way than they do now, thereby adapting them to the different needs for educational, physical [84], and mental health needs. Students should go to school to be happy, to learn, to relate, and to enjoy those who live, make, and feel with in that context. It is essential to create a life-friendly school climate that encourages meaningful and authentic learning. People cannot continue to ignore the alarming figures of childhood depression, stress, and frustration of our students. It is imperative to address this current problem in schools, which must involve prioritizing and enhancing physical and mental health. In this sense, didactic models that propose active methodologies, ones that are based on the benefits of play for learning, can make the school experience full of pleasant and constructive experiences from an emotional point of view. One of these didactic strategies is gamified physical activity as a powerful tool for achieving adequate physical and mental development in students.

## 6. Conclusions

Gamified physical exercise allows for improvements in the mental health of adolescent students as it produces effects such as well-being, fun, sociability, satisfaction, motivation, autonomy, an improvement in mood, creativity, a decreased feeling of shame, and the avoidance of depression and stress. The relationship between adolescent mental health and physical activity being mediated by gamification and exergames has been revealed. In this sense, it is concluded that the following steps are necessary:To propose that educational, health, and sport centers recommend the use of gamification methodologies for the practice of physical activity.To improve apps to achieve a long-term motivating effect that allows them to convert their use into a habit and to make them increasingly customizable and accessible for the achievement of genuinely effective results.To develop more specific apps aimed at different ages and educational stages.To take advantage of the benefits of gamified physical activity to integrate it into the day-to-day life of schools, thus favoring the practice of gamified physical activity and leisure time for the consolidation of habits and changes that promote well-being, as well as physical and mental health. In addition, further studies should be conducted on whether gaming effects mental health and in what way, and then future suggestions should be described based on them.To increase the interest of the scientific community in researching and conducting studies on the practice of physical activity and a focus on mental health, which will allow for the development of preventive intervention plans in schools.

Research on gamification and adolescent mental health needs to address multiple areas of improvement. Firstly, it is essential to strike a balance between the use of quantitative and qualitative methods. While quantitative data are commonplace, a qualitative approach should be included to gain in-depth insights into adolescent experiences.

Furthermore, consideration should be given to adapting gamified applications for diverse groups of adolescents. This involves designing interventions that are culturally sensitive and consider the individual preferences of young people.

Lastly, research should expand internationally and explore various cultural contexts. This will enable us to understand how the effectiveness of gamification in improving the mental health of adolescents may vary in different parts of the world.

In summary, enhancing research on gamification and adolescent mental health involves balancing approaches, adapting to diversity, and broadening its geographical scope.

## Figures and Tables

**Figure 1 healthcare-12-00124-f001:**
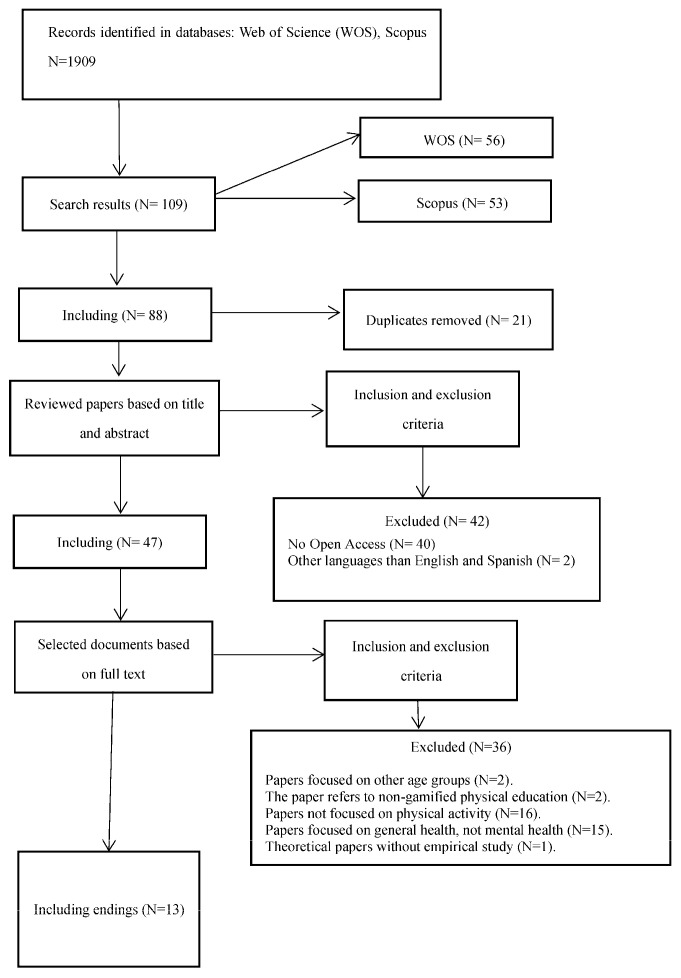
Document review flow chart.

**Table 1 healthcare-12-00124-t001:** The inclusion and exclusion criteria.

Inclusion Criteria	Exclusion Criteria
Studies in English and Spanish. Papers written from the beginning of the research on the topic to March 2023. Papers focused on samples who were aged from 10 to 19 years. Research papers. Gamification applied to emotional or psychological well-being.	Studies in languages other than English and Spanish. Documents related to other age ranges. Reflection articles. Papers that do not focus their results on gamified physical activity. Papers on emotional or psychological well-being that do not focus on gamification.

**Table 2 healthcare-12-00124-t002:** Summary of the main characteristics and results of the selected studies.

Author/Year	Objectives	Sample	Country	Method	Results
[57]	Determine which apps are useful as an educational resource in physical education classes for secondary school adolescents.	31 apps related to the practice of physical activity and the promotion of wellness.	Spain	Methodology based on the contents analysis of free apps designed for the practice of physical activity	They did not find specific applications adapted for ages between 12 and 17. There are no specific applications on the market that are suitable for the educational approach of sports content and classroom-level practice. Most applications for physical activity are mainly related to cardiovascular or strength exercise. They are intended for users of very heterogeneous ages and do not consider their characteristics. They do not have a suitable design to facilitate their didactic use. This means that they lack the educational potential needed to be used in the classroom
[58]	To analyze the impact of the gamified instrument in the subject of physical education in the educational stages from 0 to 18 years old.	17 documents	Spain	Systematic literature review	No papers were found on the infant education stage; 7 documents were found on the primary stage and 10 for adolescents from 12 to 18 years of age. The studies found improvements in motivation for learning and for the consolidation of healthy habits.
[59]	To know and assess the apps in the market that allow for working out and reducing anxiety in people.	167 anxiety apps for Android and IOS (15 on exergames)	Morocco and United Arab Emirates	Systematic literature review of apps.	Apps can serve as tools to help people suffering from general anxiety or anxiety disorders, anytime, anywhere. Apps based on gamified physical activity were shown to be effective in managing anxiety and stress states.
[60]	To advance in the field of mental health by showing that applied games are effective for the preventing of this type of problem.	9 articles on applied games and their benefits for mental health.	Australia, New Zealand, United Kingdom, and the Netherlands	Systematic literature review	The different games applied favor a better mental health. Regarding the exergames, they help to reduce depression.
[61]	Examining the cross-cutting relationship between physical activity and mental well-being.	Reference sample of 1686 people from the school/community 167 give results.	Scotland	A quantitative methodology using the Scottish Physical Activity Screening Questionnaire (Scot-PASQ) scale to assess physical activity and mental well-being with the Warwick-Edinburgh Mental Wellbeing Scale (WEMWBS).	The results provide preliminary evidence of the potential role of gamification-based physical activity in improving mental health.
[62]	Use gamification with 60% physical activity in activities and tasks performed to promote the well-being of children, preteens, and families during the pandemic.	58 parents and 82 children and pre-adolescents under 12 years old	Spain	Quasi-experimental design, with a questionnaire, qualification scales, and semi-structured interviews. Mixed methodology (qualitative and quantitative).	Families improved perceptions of their children’s physical and psychological well-being, and the results are remarkably encouraging, moving from risk scores to regular scores. These results were supported by positive feedback from parents who said it was a fun program, helped them express their emotions to the rest of the family, and helped them resume physical activity.
[63]	Create an exergame that avoids the sedentary life that can take children from 4 years of age.	8 sport professionals	Netherlands	Experiential methodology	The platform should promote children’s self-regulation and autonomy and encourage them to play sports daily.
[64]	Improving cognitive and academic performance in adolescents through the hybridization of a gamified formative model in physical education.	211 secondary school adolescents.	Spain	Development of a gamified training program in physical education. Evaluation through standardized assessment tests. Specifically, verbal fluency, planning, cognitive inhibition, and academic performance were evaluated.	The intervention showed improvement in cognitive performance, but not in academic performance. Specifically, the intervention was able to improve cognitive inhibition and verbal fluency.
[65]	Evaluating an online program to promote a healthy lifestyle and reduce the risk of eating disorders and obesity in the school setting.	10 adolescents between 14 and 18 years old.	Austria	Mixed methodology (qualitative/quantitative) with voice-over tasks, semi-structured interviews, and questionnaires.	The results of this study showed that problems with the use of and engagement and adherence to the program stem from specific issues related to the environment, such as stress at school or the possibility of using the program during school hours. For prevention interventions, intrinsic motivation plays an important role, as members of the potential target group generally do not feel a particular drive or level of distress that motivates them to use an online program
[66]	The objective of this study was to promote appropriate attitudes and behaviors in adolescents to improve their physical and psychological health.	26 primary children, 28 secondary school adolescents, 29 high school adolescents, and 9 university adolescents	Spain	The methodology was qualitative; they created a gamified app in collaboration with the students through questionnaires that later evaluated it.	An app was obtained to improve the mental and physical health of adolescents. This application had great didactic and practical value and was certified as a healthy application by the Health Quality Agency of Andalusia (ACSA).
[67]	To know the benefits for the mental health of primary school adolescents of the gamified game and the “exergame”.	417 elementary school adolescents and 8 teachers from 4 schools.	Spain	Qualitative methodology. Observation and records (field notes), questionnaires, individual interviews and discussion groups were used.	It was shown that the intervention carried out through gamification and the “exergame” produced positive changes in the mood and well-being of the adolescents. They had fun, were highly motivated, and creativity and enjoyment in what they were learning was encouraged. They decreased their feelings of embarrassment, showed interest and motivation for the dance, and learned in an autonomous way.
[68]	To know the benefits and problems of gamified e-health apps.	46 articles on gamified e-health apps.	Morocco and Spain	Systematic literature review	Gamification is shown in health and wellness contexts related to chronic diseases, physical activity, and mental health. Application reward systems were confirmed to be effective in motivating and generating short-term engagement among users.
[69]	Understand the impact of gamification interventions on m-Health apps to improve physical activity levels.	50 papers about m-Health apps	China and Belgium	Systematic literature review	Effectiveness of gamification to improve participation in physical activity.

## Data Availability

The authors will provide more information to the interested parties through the corresponding author.

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
