# Peer review of "The Relationship between Gamified Physical Exercise and Mental Health in Adolescence: An Example of Open Innovation in Gamified Learning"

_healthcare, 2024, doi:10.3390/healthcare12020124_

Round 1
Reviewer 1 Report
Comments and Suggestions for Authors
The authors have provided important information regarding the advantages of gamified physical exercise and mental health.
However, the reviewer believes that several changes would be required to improve the quality of this manuscript.
Here are several of my comments:
1. Please proofread the manuscript for grammatical and writing errors.
2. Please re-structure the manuscript. The current version of the manuscript is very hard to follow. For example, the authors could start the review by describing what are gaming exercise, what is mental heatlh , and then the connection between both.
3. Most of the paragraphs only consist of one sentence, this should be revised. paragraphs normally consist of minimal 3 sentences.
4. The introduction is too long.
5. Please include in the method section, which databases were used for searching of articles, was it only from scopus and WOS? Why only used these two sources?
6. There is an empty bullet block in Figure 1. Please revise.
7. Is there any criteria of what is the oldest year of the paper included in this review? the inclusion criteria only mention until April 2023.
8. This review lacks a comprehensive and strong conclusion: please describe based on literature does gaming effect mental health and in what way. then describe future suggestions based on that.
Comments on the Quality of English LanguageThe English requires minor editing.
Author Response
Dear reviewer, thank you very much for the recommendations received. We have addressed the suggested changes and edited the manuscript by the editing service of MDPI.
You will see the changes in the main text where the deleted text is crossed out in red and the added text in blue.
We attach the letter of reply to the comments and certificate of English edition

Reviewer 2 Report
Comments and Suggestions for Authors
Pls see attached file.

Pls see attached file.
Author Response

(The authors gave the same response as above.)

Reviewer 3 Report
Comments and Suggestions for Authors
Please move the process of extracting the contents of the literature search and the papers used for analysis at the beginning of the result to the research method.
p7 4. discussion –> result
Although the discussion is well organized, there is a regret that the content of the discussion will be enriched. I hope that the contents of the process of comparative analysis with the results of other previous studies will be added.
5. In the conclusion, please add the limitations shown in the study and future research suggestions.
.
Author Response

(The authors gave the same response as above.)

Round 2
Reviewer 1 Report
Comments and Suggestions for Authors
The authors have reviewed an important topic of how gamified physical exercise affects mental health.
Line 39 - Please describe, what type of disease are the authors referring to.
Line 45 - What type of sports?
Line 48 - What type of platforms and apps?
Line 51 - What type of habits and behaviours ?
Line 54 - What do you mean by "funny" way? would there be a better way to describe it?
Line 75 - "The study of [1] refers to mental health problems " ... such as???
Line 94 - Please describe what a sedentary lifestyle is?
Line 153 - Please specify the range of the publication years that was considered in this study.
Table 1 - Please specify when was the begining of the research topic.
Line 255 - the authors mentioned that the search was carried out between december - march 2023, while in line 153, the authors mentioned all studies written to date, which implys the search was done on all articles even the very old ones. Please clarify which is the right year range for the article search used in this review.
- Please mention in the method section, how many authors has performed and read the papers that are choosen to be used in this review.
In general, I believe the manuscript requires more editing in the writing and structure to be more understandable. For example, paragraph one in the introduction (Line 27) is hard to follow. Moreover, it would be better to describe things in more detail.
Comments on the Quality of English Language
Moderate editing of English language required.
Author Response
You can see in the attached document the Response to the Reviewer’s Comments
Regards

Reviewer 2 Report
Comments and Suggestions for Authors
Accept
Author Response
Dear reviewer, thank you for your feedback and for finally accepting the manuscript
Regards